# Latent Tuberculosis Infection among Healthcare Students and Postgraduates in a Mediterranean Italian Area: What Correlation with Work Exposure?

**DOI:** 10.3390/ijerph17010137

**Published:** 2019-12-24

**Authors:** Maria Gabriella Verso, Nicola Serra, Antonina Ciccarello, Benedetta Romanin, Paola Di Carlo

**Affiliations:** 1Occupational Health Unit, Department of Health Promotion Sciences, Maternal and Infant Care, Internal Medicine and Medical Specialties “G. D’Alessandro”, University of Palermo, via del Vespro 143, 90127 Palermo, Italy; 2Statistics Unit—Department of Public Health, University Federico II of Naples, via Sergio Pansini, 5, 80131 Naples, Italy; nicola.serra@unina.it; 3School of Specialization in Occupational Health, Department of Health Promotion Sciences, Maternal and Infant Care, Internal Medicine and Medical Specialties “G. D’Alessandro”, University of Palermo, via del Vespro 143, 90127 Palermo, Italy; antonina.ciccarello@you.unipa.it; 4School of Specialization in Infectious Diseases, Department of Health Promotion Sciences, Maternal and Infant Care, Internal Medicine and Medical Specialties “G. D’Alessandro”, University of Palermo, via del Vespro 129, 90127 Palermo, Italy; benedetta.romanin@gmail.com; 5Infectious Diseases Unit, Department of Health Promotion Sciences, Maternal and Infant Care, Internal Medicine and Medical Specialties “G. D’Alessandro”, University of Palermo, via del Vespro 129, 90127 Palermo, Italy; paola.dicarlo@unipa.it

**Keywords:** medical and nursing students, occupational biohazard, Quantiferon-TB test, latent TB infection, Mantoux skin test

## Abstract

Background: Tuberculosis screening is part of the standard protocol for evaluating the risk of infection in healthcare workers. The aim of this study was to evaluate the prevalence of latent tuberculosis infection (LTBI) among students attending various healthcare profession degree courses and postgraduate medical courses at the School of Medicine of the University of Palermo, Italy, and assess the possible professional origin of infection. Methods: In total, 2946 students (2082 undergraduates and 864 postgraduates) took part in a screening program for LTBI between January 2014 to April 2019 using the tuberculin skin test (TST). Students with a positive TST result underwent a Quantiferon-TB test (QFT). Results: Among the 2082 undergraduates, 23 (1.1%) had a positive TST; the result was confirmed with QFT for 13 (0.62%) of them. Among the 864 postgraduate students, 24 (2.78%) had a positive TST and only 18 (2.08%) showed a positive QTF. Latent tuberculosis infections were significantly more frequent among postgraduates than undergraduates (2.08% > 0.62%, *p* < 0.0001). There was a higher number of subjects previously vaccinated for TB (18.87% > 0.24%, *p* < 0.0001), and of vaccinated subjects found positive for TST and QTF (66.67% > 7.69%, *p* = 0.001) in the postgraduate group. Conclusion: Latent TB is relatively low among medical school students in our geographic area. Nevertheless, this infectious disease must be regarded as a re-emerging biohazard for which preventive strategies are required to limit the risk of infection, especially among exposed workers.

## 1. Introduction

Italy is one of the European countries with a low incidence of tuberculosis. ‘Tuberculosis surveillance and monitoring in Europe 2019′ showed a notification rate of 6.5 per 100,000 population in Italy in 2017, with a mean annual change in the notification rate of −0.6%. The report showed a TB mortality rate of 0.6 per 100,000 population in Italy in 2017 [1].

Analysis by Italian age group showed that the rate was 15 per 100,000 for 15- to 24-year-olds and 9.4 per 100,000 for 25- to 44-year-olds. Italy was the birthplace of 33.8% of TB cases, and the remaining cases were among the foreign population [1].

Due to geographical location, some EU areas like Sicily are today facing epochal inflows of new economic migrants and asylum seekers. Tuberculosis is a frequent cause of hospitalization among migrants coming from areas where circulation of *Mycobacterium tuberculosis* is high [2].

Tuberculosis is a preventable disease. Identifying latent TB infection (LTBI) is vital to achieving the goal of controlling and eliminating TB with specific therapeutic measures.

Although there is no diagnostic gold standard for LTBI, two tests are available to identify it: The tuberculin skin test (TST) and the gamma interferon (IFN-γ) release assay (IGRA) [3]. Diagnosis of LTBI via the tuberculin skin test (TST) remains the conventional and most used screening test in Italy. Screening should be considered for risk groups in order to promptly detect and manage this condition.

Compared to the general population, healthcare workers (HCWs) are known to be at an increased risk of acquiring tuberculosis (TB). Surveillance of other professional categories, such as the Italian State Police (ISP) and undergraduate and postgraduate students attending medical school, has been described and outbreaks of TB disease reported [4,5,6,7,8].

According to Italian law, students receiving practical training in the territory’s hospitals are regarded as workers exposed to biohazards, as are physicians and nurses regularly employed and working in the various departments. Therefore, they undergo regular health surveillance [9]. 

To evaluate the epidemiology of LTBI in undergraduate and postgraduate Sicilian students attending medical school, we conducted a surveillance study on the usefulness of identifying LTBI-positive subjects and risk factors among healthcare students in Palermo, southern Italy, using TST, and subsequently Quantiferon testing (QFT-IT) for those with a positive TST result. 

The main health surveillance system for workers exposed to contagious TB involves monitoring latent tuberculosis infection by:-Identifying infected subjects to prevent evolution from infection to disease;-Reviewing the adequacy of the protocols; and-Reclassifying risk levels.

According to the guidelines of the Italian Ministry of Health, published in 2009, the recommended procedure is based on the tuberculin test (TST) [10]. The procedure can be integrated with the Quantiferon TB test (QFT-IT) to confirm a positive TST result. Exclusive use of QFT-IT, if available, can be considered if the group in question has high rates of BCG vaccination or if high rates of positive TST (Mantoux test) results are expected. The aim of this study was to evaluate latent tuberculosis infection in asymptomatic students and assess the possible professional origin of infection.

## 2. Materials and Methods 

### 2.1. Study Design 

This was a cross-sectional study, including all students attending healthcare profession and postgraduate medical courses at the School of Medicine of the University of Palermo, Italy. 

LTBI infection was defined by a positive tuberculin skin test (TST) result followed by a confirmatory positive Quantiferon TB test result in an asymptomatic subject [11].

### 2.2. Sampling and Eligibility 

The inclusion criteria for this study were: Asymptomatic undergraduate and postgraduate students, who gave informed consent. Exclusion criteria included: Knowledge of a previous positive reaction to the Mantoux test or to QTF-IT; lost to follow-up due to missed scheduled appointments to collect data, such as TST testing, quantiferon analysis, and clinical record; inaccurate data provided; and refusal to sign informed consent. None of the medical school students were disqualified for exclusion criteria, and all those who presented for the medical examination were enrolled. 

The study was performed at the University of Palermo’s Department of Health Promotion Sciences, Maternal and Infant Care, Internal Medicine, and Medical Specialties “G. D’Alessandro”, and involved a sample of 2946 students enrolled between January 2014 and April 2019.

The study population was made up of:2082 undergraduate medical school students attending various degree courses (UMSs), 39.35% male and 60.65% female, aged 18 to 64 years old, with a mean age of 23.80 years old, and standard deviation of 5.01 years old;864 postgraduate medical students (PMSs), 42.36% male and 57.64% female, aged 25 to 57 years old, with a mean age of 30.12 years old, and standard deviation of 3.15 years old.

We compiled a clinical record for each student, which included information on any conditions or diseases affecting family members, and personal remote and proximate pathologies (anamnesis). We then conducted a physical examination and Mantoux test. HIV and HCV status were analyzed, with none of the participants testing positive. All the enrolled subjects were vaccinated against HBV. All the participants were asked to sign an informed consent letter prior to the processing of data. The study was approved by the local Ethics Committee—Palermo 1.

Participant anonymity was guaranteed. No economic incentives were offered or provided for participation in this study. 

Using anamnesis data, we collected information on the following variables: Age, sex, nationality, incidence of the disease in the country of origin (low/medium/high), parental history of TB, travel or work in countries with a high incidence of TB (not performed by any of them), degree or specialization and year, specifying if the student had already carried out in-hospital training courses (clinical) or not (preclinical), and any previous vaccination against TB (BCG vaccination). 

The tuberculin skin test (TST) was performed by trained personnel following standard procedures: 0.1 mL of purified protein derivative (Tubertest^®^; Sanofi Pasteur, Lyon, France) was injected into each participant [10]. The TST was administered to the volar side of the participant’s forearm and read 48 to 72 h after application, according to the guidelines of the Italian Ministry of Health, published in 2009 [10]. A positive TST is defined as an induration measuring ≥10 mm in healthy subjects regardless of BCG history [11]. A diameter of 5 mm or more was considered positive in HCWs after accidental contact with infectious patients or materials (identified as “contacts”).

All those with positive TST results were tested two months later with an Interferon-Gamma Release Assay (IGRA; Quantiferon TB-Gold Cellestis, Victoria, Australia) to confirm LTBI diagnosis, due to the latter’s greater specificity compared to conventional TST. 

Several studies have shown how IGRA is more specific than TST. A false positive TST result can be seen, for example, in people who have previously received the BCG vaccine, or in patients with autoimmune diseases [12,13,14]. Various studies have confirmed the role of QFT-IT in reducing the potential occurrence of “false positive” results due to exposure to atypical *Mycobacteria* or BCG vaccination [15,16].

QFT-IT was performed using TB antigen-, mitogen, and unstimulated tube (nil) at the Central Laboratory of Advanced Diagnosis and Biomedical Research (CLADIBIOR), University Hospital “P. Giaccone”, Department of Biopathology and Medical Biotechnology, University of Palermo, Italy, as previous reported [17].

The cutoff value of the Quantiferon TB test is 0.35 IU/mL greater than the null control for the plasma sample stimulated with the TB antigen: An IFN-γ ≥ 0.35 IU/mL (TB antigens minus negative control) was considered a positive test. Subjects with an antibody TB response above the cutoff rate are probably infected.

The IFN-γ concentration level is not necessarily correlated with the stage or degree of infection, or the probability of disease progression, but can be influenced by the subjects’ immune status. Cases of patients with cavitary tuberculosis, cancer, or HIV infection showing low IFN-γ concentration levels have been described in the literature [18,19,20]. A positive test does not necessarily indicate the presence of active tuberculosis.

Students with a diagnosis of latent tuberculosis were examined by an infectious diseases specialist and underwent a chest X-ray to exclude active TB disease. In the absence of clinical and radiographic signs of active TB, they were considered to have LTBI and underwent a clinical follow up at 6, 12, and 24 months after diagnosis. An infectious diseases consultant evaluated all LTBI cases and the decision to prescribe prophylaxis was made on a case-by-case basis. 

### 2.3. Statistical Analysis

Statistical analysis was performed using the Matrix Laboratory (MATLAB) analytical toolbox version 2008 (MathWorks, Natick, MA, USA). Data are presented as the number and percentage for categorical variables, and continuous data expressed as the mean ± standard deviation (SD) unless otherwise specified. The χ^2^ test and χ^2^ test with Yates’s continuity correction were performed to evaluate significant differences in proportions or percentages between the two groups. Particularly, the χ^2^ test with Yates’s continuity correction was used where the χ^2^ test was not appropriate. In addition, the Binomial test was performed to compare two mutually exclusive proportions or percentages. A significant difference between two means was evaluated by the Student’s *t*-test. The multivariate analysis was performed with the multiple comparison chi-square test, to define significant differences among percentages (more than two). In this case, if the multiple chi-square test was significant (*p*-value < 0.05), the post hoc test (Z-test) was performed to identify the significant highest and lowest percentages. Finally, all tests with *p*-value (*p*) < 0.05 were considered significant. 

## 3. Results

Table 1 shows the general characteristics of the undergraduate medical students (UMSs = 2082 subjects) and reports parameters, such as gender, nationality, their country’s prevalence of tuberculosis (low and high), university training completed (preclinical or clinical depending on whether they had already been in contact with patients), and TB vaccination. Subjects were divided into students with a negative TST test (Mantoux), those with a positive TST test only, and those who tested positive to both TST and QFT-IT. In addition, we report the various medical school courses the participants attended. The percentages of students with or without a positive LTBI test result were determined considering the total number of students enrolled in each course. Only 0.14% (3/2082) had received BCG vaccination. In addition, 1.10% (23/2082) had a positive Mantoux test result (two had received BCG vaccination), and 56.52% (13/23) of these subjects also had a positive QFT-IT result (one case of BCG vaccination).

Table 1 highlights a significant presence of females (F) compared to males (M) (F: 64.65% (1346/2082) > M: 35.35% (736/2082), *p* < 0.0001), and there was a significant gender difference as regards students who were LTBI positive for QFT-IT (F: 69.23% > M: 30.77%, *p* = 0.0027). In this case, the significant number of females who were LTBI positive for QFT-IT might be conditioned by the significant presence of females in the UMS group.

Similar to Table 1, in Table 2, we report the characteristics of the postgraduate medical students (PMSs = 864 subjects). In this case, the postgraduate courses are grouped into three areas: Medical (total 398), surgery (total 189), and assistance (total 277). In total, 18.87% of the group participants (163/864) had received BCG vaccination. It should be remembered that up until 2001, TB vaccination was mandatory in Italy for students on degree courses in medicine [21,22]. In total, 2.78% (24/864) had a positive Mantoux test; 15 of those 24 had received TB vaccination (62.5%). In total, 75% (18/24) of the students with a positive Mantoux test result had a positive QFT-IT test; 66.67% (12/18) of those with a positive QFT-IT test result had also been vaccinated against TB. Regarding the three course area groups, among the 2.78% (24/864) postgraduate students with a positive Mantoux test result, 2.01% (8/398) were in the medical area group, 3.70% (7/189) in the surgical area group, and 3.25% (9/277) in the assistance area group.

Table 2 shows a significant presence of females (F) compared to males (M) (F: 57.64% (498/864) > M: 42.36% (366/864), *p* < 0.0001), but there was no significant gender difference as far as subjects who tested positive for QFT-IT was concerned (F: 61.11% > M: 38.89%, *p* = 0.053). In this case, we observed a significant presence of males in the PMS group compared to the UMS group (42.36% > 35.35%, *p* = 0.0003), and consequently, the greater male presence in the PMS group implies there was no significant difference between females and males who tested positive for QFT-IT.

Regarding nationality, although the sample consisted almost exclusively of Italians, with only eight foreigners, we found in the UMS group one positive student from a high incidence country (Ukraine), which he had visited before his hospital training, while none of the foreigners in the PMS group were positive.

The foreign students in the sample came from Greece (one postgraduate student), Morocco (two undergraduate students), Ukraine (two undergraduate students and one postgraduate), Madagascar (one undergraduate student), and Sri Lanka (one undergraduate student); in all these countries, apart from Greece, the prevalence of TB infection is high.

In Table 3, we present all the medical specialization courses and the percentage of postgraduates with positive TB results by both the Mantoux and QFT-IT test.

Table 4 shows five different sections of statistical tests performed in this study: (1) On students without LTBI versus LTBI-positive students by both Mantoux and QFT-IT; (2) on LTBI-positive students by both Mantoux and QFT-IT in the UMS group; (3) on LTBI-positive students by both Mantoux and QFT-IT in the PMS group; (4) the UMS group versus the PMS group; and (5) the UMS group versus the PMS group as regards students with LTBI.

In the first section of Table 4, we can see that in the PMS group, there is a significant difference regarding the age of students without LTBI and those who tested positive for LTBI by both Mantoux and Quantiferon: 30.06 years old < 32.56 years old (*p* = 0.0008), i.e., subjects with LTBI were older than students without LTBI.

As far as university training is concerned, no significant differences were observed in the UMS or the PMS group, or between students without LTBI and students with LTBI in both the clinical (UMS: 61.92% > 53.85%, *p* = 0.756; PMS: 100% = 100%, *p* < 0.0001) and preclinical subgroups (UMS: 38.08% < 46.15%, *p* = 0.756; PMS: 0.0% = 0.0%, *p* < 0.0001). 

BCG vaccination analysis showed a higher percentage in both the UMS and the PMS group of BCG-vaccinated students with LTBI compared to students without latent tuberculosis (UMS: 7.69% > 0.15%, *p* = 0.0026; PMS: 66.67% > 17.62%, *p* < 0.0001, respectively). 

In the second section of Table 4, we report the characteristics of students in the UMS group with LTBI. This analysis showed a significant presence of females vs. males (69.23% > 30.77%, *p* = 0.0027), and a significant presence of students without BCG vaccination vs. students with BCG vaccination (92.31% > 7.69%, *p* < 0.0001).

In the third section of Table 4, we report the characteristics of the students in the PMS group with LTBI. This analysis showed no significant presence of females (61.11% > 38.89%, *p* = 0.053), but a significant presence of students with clinical university training (100%, *p* < 0.0001). 

Finally, we observed a significant connection between all the postgraduate schools and LTBI (*p* = 0.0042). Post hoc analysis (Z-test) showed that the medical schools with a significantly higher number of students with a positive Mantoux test, confirmed at QFT-IT, were: Digestive system surgery (50%, *p* < 0.0001), emergency surgery (50%, *p* < 0.0001), urology (83.33%, *p* = 0.0147), and otolaryngology (86.67%, *p* = 0.0457). 

In the fourth section of Table 4, we compare the general characteristics of the UMS and PMS groups. Compared to the UMS group, there was a significant presence in the PMS group of males (42.36% > 35.35%, *p* = 0.0003), students with clinical university training (100% > 61.96%, *p* < 0.0001), vaccinated students (18.87% > 0.24%, *p* < 0.0001), and students with positive Mantoux and QFT-IT results (2.08% > 0.62%, *p* < 0.0001), while in the UMS group, there was a significant presence only of students with preclinical university training (39.04% > 0.00%, *p* < 0.0001).

Finally, in the fifth section, we compare the UMS and PMS groups, considering students with LTBI only. Significant differences were found only for university training (UMS preclinical: 46.15% > PMS preclinical: 0.0%, *p* = 0.0060; UMS clinical: 53.85% < PMS clinical: 100%, *p* = 0.0060) and TB vaccination (UMS: 7.69 < PMS: 66.67%, *p* = 0.001). 

We found a statistically significant result regarding the higher number of students with LTBI in the PMS group compared to the UMS group (2.08%, >0.62%, *p* < 0.0001), as described in Table 4.

This corresponds with the former’s increased exposure in terms of weekly contact hours with patients, and a higher risk of exposure in the workplace to patients with active TB.

We met again with the students who had a positive skin test result and asked them if they remembered having had contact with patients with TB in the workplace or in other contexts: None of the students knew how or when they had become infected.

## 4. Discussion

The analysis of the results led us to make the following considerations. 

Of the 2946 enrolled subjects, the number that were found to be LTBI positive with the Mantoux test was 47 (23 in the UMS group and 24 in the PMS group), and for 31 of them, diagnosis was confirmed with QFT-IT (13 UMS and 18 PMS). 

A large part of the students enrolled in our study were Italian, and Italy is considered to be one of the European countries with a low incidence of tuberculosis [23]. 

Only in the UMS group was there a significantly higher presence of females with LTBI. These results are in accordance with recent data, which showed that 59% (in OECD countries) and 66% (in Italy) of PhD graduates in the healthcare field are women [24]. In this case, the significant number of females who were found to be LTBI positive with QFT-IT could be conditioned by the significant presence of females in the UMS group. In the PMS group, there was no significant statistical gender difference as regards the incidence of latent tuberculosis infection.

Most of the enrolled subjects were undergraduate students who had had preclinical and clinical university training; on the other hand, all the postgraduate medical students had clinical experience in a hospital setting. There was a significantly higher number of students with LTBI in the PMS group, particularly among older subjects. This may be due to their longer clinical training in terms of exposure to patients, and greater risk of exposure to occupational infections.

TB vaccination analysis showed a significant difference between the two groups of students: 17% of postgraduate medical students had received BCG vaccination versus 0.24% of undergraduate students. 

As previously mentioned, compulsory BCG vaccination for all students attending medical faculty degree courses in Italy was abolished in 2001, and the vaccine is now only reserved for students exposed to multi-drug-resistant *Mycobacteria*. Thus, this undergraduate group was excluded from tuberculosis (TB) vaccination [22]. On the contrary, as some of the postgraduates had started their studies before 2001, their group showed a higher rate of BCG immunization. 

Quantiferon analysis for Mantoux-positive students reduced the number of them with latent tuberculosis, since the test can be influenced by various factors, including skin hyperreactivity or BCG vaccination. In this regard, it is interesting to observe how some vaccinated subjects in our sample are infected with *Mycobacterium tuberculosis*: (US: 7.69 < PMS: 66.67%). This confirms that BCG vaccination is not always effective in preventing *Mycobacterium tuberculosis* infection. The benefits of BCG have been vehemently debated for many years and a consensus has been hard to reach, especially as far as the adult population is concerned. 

Regarding the specialization schools attended by our participants, no significant statistical differences between the various areas have emerged, although the number of subjects with LTBI is very small. 

We observed that digestive system surgery, emergency surgery, urology, and otolaryngology were the specialization schools with a significantly higher presence of students with LTBI, although the number of students in these schools was low.

As regards the fact that some of our subjects with LTBI attended specialization schools that are not historically linked to TB, such as digestive system surgery, emergency surgery, urology, and otolaryngology, we believe that students in these contexts have still not developed the critical thinking required to foresee the importance of taking universal precautions. Thus, they tend to consider the use of gloves to be adequate self-protection and fail to take any prevention measures against tuberculosis. This attitude is reported by Bergamini et al. [25]. Similar studies have also found LTBI in health workers in unexpected departments, such as psychiatry [26]. Moreover, the general surgery setting had the highest percentage of LTBI, as well as other injuries, among healthcare workers in southern Italy [27]. 

Due to the current very frequent migration from countries with a high incidence of tuberculosis to Italy, it is likely that the incidence of LTBI and TB disease will increase in the general population.

In this regard, faculty of medicine students and, even more so, specialists and doctors in service in hospitals will be at a greater risk of contracting infection and illness at work.

As also highlighted in this study, the risk is not confined to traditional specialist contexts, such as the infectious disease units, emergency departments, radiology, or pneumology. It also exists in very different specialist contexts, where patients, sometimes without a diagnosis of tubercular disease, present with other coexisting diseases. Greater attention is needed when working in contexts where the risk exists. Moreover, healthcare workers in Italian companies are required by law to participate in training and information programs.

The incidence of LTBI among the medical students in our study turned out to be very low, similar to what has been observed in analogous contexts in other Italian universities, but lower than in recent German studies, and is work-related [7,28,29,30].

## 5. Conclusions

Although this study detected just a few cases of LTBI, its prevalence in our population may be relevant to improving our knowledge of latent tuberculosis among Italian young adults, especially those exposed in the workplace, as data relating to LTBI in the general Italian population are not known. Our results confirm that the incidence of LTBI is low in Italy but also that effective prevention strategies need to be implemented in the university hospital under study. 

However, the authors stress that, as in the case of healthcare workers (HCWs), there is an occupational risk of students undergoing clinical training contracting tuberculosis, due to close and prolonged contact with patients.

In a country like Italy, where the presence of TB is low, screening programs for healthcare students may be useful in this era of international exchange for early identification and treatment of LTBI [31,32,33].

### 5.1. Limitations

This study had a good sample size but it was conducted in a single geographical area. Therefore, a multicenter study that examines different geographical areas is needed in order to determine the prevalence and incidence of latent tuberculosis infection and associated risk factors among HCWs student, and their adherence to LTBI screening.

### 5.2. Declarations

*Ethics approval and consent to participate:* Informed consent was signed by all participants in the study. Anonymity was guaranteed for all participants. The study was approved by the Local Ethics Committee Palermo-1 on 23 May 2016 (document n. 21).

## Figures and Tables

**Table 1 ijerph-17-00137-t001:** Characteristics of 2082 medical school student (MSS) participants, stratified into medical school students with and without latent tuberculosis (LTBI).

	Undergraduate Medical Students (UMS) (Mean ± SD/Percentage/Number)		
Parameters	Students without LTBI	Students with Positive Mantoux Test	Students with Positive QFT-IT Test *
Total	98.90% (2059/2082)	1.10% (23/2082)	56.52% (13/23)
*Age* (y.o.)	23.77 ± 4.97	26.59 ± 7.13	26.36 ± 6.07
*Gender*			
Male	35.36% (728/2059)	34.78% (8/23)	30.77% (4/13)
Female	64.64% (1331/2059)	65.22% (15/23)	69.23% (9/13)
*Nationality*			
Italian	99.81% (2055/2059)	91.30% (21/23)	92.31% (12/13)
Other	0.19% (4/2059)	8.70% (2/23)	7.69% (1/13)
*Medical School courses*			
Medical Biotechnology	100% (3/3)	−	−
Speech therapy	100% (31/31)	−	−
Medical Radiology techniques	100% (35/35)	−	−
Physiotherapy	97.44% (38/39)	2.56% (1/39)	−
Obstetrics	100% (43/43)	−	−
Healthcare professional	97.26% (71/73)	2.74% (2/73)	2.74% (2/73)
Dentistry	100% (111/111)	−	−
LM in Nursing/Obstetrics	100% (145/145)	−	−
Medicine and Surgery	98.06% (455/464)	1.94% (9/464)	1.08% (5/464)
Nursing	99.03% (1127/1138)	0.97% (11/1138)	0.53% (6/1138)
*Country incidence of TB*			
Low	99.81% (2055/2059)	91.30% (21/23)	92.31% (12/13)
High	0.19% (4/2059)	8.70% (2/23)	7.69% (1/13)
*University training*			
Preclinical	38.08% (784/2059)	34.78% (8/23)	46.15% (6/13)
Clinical	61.92% (1275/2059)	65.22% (15/23)	53.85% (7/13)
*TB Vaccination*			
Yes	0.15% (3/2059)	8.70% (2/23)	7.69% (1/13)
No	99.85% (2056/2059)	91.30% (21/23)	92.31% (12/13)

QFT-IT test = Quantiferon test; * = The quantiferon test was performed on subjects with a positive Mantoux test result.

**Table 2 ijerph-17-00137-t002:** Characteristics of 864 postgraduate medical students (PMS) participants, stratified into postgraduate medical students with and without latent tuberculosis (LTBI).

	Postgraduate Medical Students (PMS) (Mean ± SD/Percentage/Number)		
Parameters	Students without LTBI	Students with Positive Mantoux Test	Students with TB Positive QFT-IT Test *
Total	97.22% (840/864)	2.78% (24/864)	75% (18/24)
*Age* (y.o.)	30.06 ± 3.13	32.29 ± 3.01	32.56 ± 3.18
*Gender*			
Male	42.50% (357/840)	37.5% (9/24)	38.89% (7/18)
Female	57.50% (483/840)	62.5% (15/24)	61.11% (11/18)
*Nationality*			
Italian	99.76% (838/840)	100% (24/24)	100%% (18/18)
Other	0.24% (2/840)	0.00% (0/24)	0.00% (0/18)
*Resident schools*			
Medical Area	97.99% (390/398)	2.01% (8/398)	1.01% (4/398)
Surgical Area	96.30% (182/189)	3.70% (7/189)	3.17% (6/189)
Assistance Area	96.75% (268/277)	3.25% (9/277)	2.89% (8/277)
*Country incidence of TB*			
Low	99.88% (839/840)	100% (24/24)	100% (18/18)
High	0.12% (1/840)	−	−
*University training*			
Preclinical	0.00% (0/840)	−	−
Clinical	100% (840/840)	100% (24/24)	100% (18/18)
*TB Vaccination*			
Yes	17.62% (148/840)	62.50% (15/24)	66.67% (12/18)
No	82.38% (692/840)	37.50% (9/24)	33.33% (6/18)

QFT-IT test = Quantiferon test; * = The quantiferon test was performed on subjects with a positive Mantoux test result.

**Table 3 ijerph-17-00137-t003:** Postgraduate students with LTBI for all medical specialization schools.

Medical Specialization Schools	Postgraduate Medical Students (Percentage/Number)	% Students with Positive Mantoux Test	% Students with Positive QFT-IT Test
*Total*	864	24	18
***Medical Area (total 398)***			
Physical and Rehabilitative Medicine	0.35% (3)	−	−
Child Neuropsychiatry	0.35% (3)	−	−
Cardioangiology	0.69% (6)		
Nephrology	0.69% (6)	−	−
Angiology	0.93% (8)	−	−
Hematology	1.16% (10)	−	−
Radiotherapy	1.16% (10)	10% (1/10)	10% (1/10)
Endocrinology	1.50% (13)	−	−
Neurology	4.17% (36)	−	−
Dermatology	1.97% (17)	−	−
Geriatrics	2.43% (21)	9.52% (2/21)	4.76% (1/21)
Pneumology	2.43% (21)	−	−
Psychiatry	2.66% (23)	−	−
Physiatrics	2.78% (24)	−	−
Medical Oncology	3.01% (26)	−	−
Internal Medicine	3.59% (31)	−	−
Gastroenterology	3.82% (33)	6.06% (2/33)	−
Cardiology	4.17% (36)	−	−
Pediatrics	8.22% (71)	4.23% (3/71)	2.82% (2/71)
***Surgical Area (total 189)***			
Mini-invasive Surgery	0.12% (1)	−	−
Pediatric Surgery	0.12% (1)	−	−
Vascular Surgery	0.12% (1)	−	−
Neurosurgery	0.12% (1)	−	−
Dentistry	0.12% (1)	−	−
Pediatric Dentistry	0.12% (1)	−	−
Digestive System Surgery	0.23% (2)	50% (1/2)	50% (1/2)
Emergency Surgery	0.23% (2)	50% (1/2)	50% (1/2)
Thoracic Surgery	0.58% (5)	−	−
Cardiac Surgery	0.69% (6)		
General Surgery	0.93% (8)	−	−
Oncological Surgery	1.16% (10)	−	−
Urology	1.39% (12)	16.67% (2/12)	16.67% (2/12)
Plastic Surgery	1.62% (14)	−	−
Ophthalmology	1.62% (14)	−	−
Otolaryngology	1.74% (15)	13.33% (2/15)	13.33% (2/15)
Orthopedics and Traumatology	3.59% (31)	4.76% (1/31)	−
Gynecology and Obstetrics	7.41% (64)	−	−
***Assistance Area (total 277)***			
Pathological Anatomy	0.69% (6)	−	−
Legal Medicine	1.27% (11)	9.09% (1/11)	9.09% (1/11)
Hygiene and Preventive Medicine	1.85% (16)	6.25% (1/16)	−
Occupational Medicine	1.85% (16)	−	−
Microbiology and Virology	2.08% (18)	−	−
Clinical Pathology	2.20% (19)	5.26% (1/19)	5.26% (1/19)
Radiology	10.76% (93)	2.15% (2/93)	2.15% (2/93)
Anesthesia and Resuscitation	11.34% (98)	4.08% (4/98)	4.08% (4/98)

QFT-IT test = Quantiferon test; * = The quantiferon test was performed on subjects with positive Mantoux test.

**Table 4 ijerph-17-00137-t004:** Statistical tests.

**Parameters**	**Section 1** **Students without LTBI vs. Students with LTBI at QFT-IT (+)**	***p*-value (Statistical test)**
*Age*	*UMS*: 23.77y.o. < 26.36y.o.	0.062 (T)
	*PMS*: 30.06y.o. < 32.56y.o.	0.0008 * (T)
*Gender (M)*	*UMS*: 35.36% > 30.77%	0.96 (CY)
	*PMS*: 42.5% > 38.89%	0.76 (CY)
*University training*		
Preclinical	*UMS:* 38.08% < 46.15% *PMS:* 0.0% = 0.0%	0.756 (CY) <0.0001 * (CY)
Clinical	*UMS:* 61.92% > 53.85% *PMS:* 100% = 100%	0.756 (CY) <0.0001 * (CY)
*TB Vaccination (yes)*	*UMS:* 0.15% < 7.69% *PMS:* 17.62% < 66.67%	0.0026 * (CY) <0.0001* (CY)
**Parameters**	**Section 2** **Analysis within UMS group ** **with LTBI at QFT-IT (+)**	***p*-value(Statistical test)**
*Gender*	69.23%(Female) > 30.77%(Male)	0.0027 * (B)
*University training*	53.85%(Preclinical) > 46.15%(Clinical)	0.58 (B)
*TB Vaccination*	92.31% (no) > 7.69% (yes)	<0.0001 * (B)
*Medical School courses*+	Healthcare professional (2.74%); Medicine and Surgery (1.08%); Nursing (0.53%)	0.223 (Cm)
**Parameters**	**Section 3** **Analysis within PMS group** **with LTBI at QFT-IT (+)**	***p*-value (Statistical test)**
*Gender*	61.11%(Female) > 38.89%(Male)	0.053 (B)
*University training*	0.00% (Preclinical < 100%(Clinical)	<0.0001 * (B)
*TB Vaccination*	66.67% (yes) > 33.33% (no)	0.0027 * (B)
*Postgraduate Schools grouped*	Medical area (22.22%); Surgical area (33.33%); Assistance area (44.44%)	0.290 (Cm)
*All Postgraduate Medical Schools*	All Postgraduate Schools Medical with frequency different to zero in Table 4 were considered	0.0042 * (Cm) Digestive System Surgery (-) **, *p* < 0.0001 (Z) Emergency Surgery (-) **, *p* < 0.0001 (Z) Urology (-) **, *p* = 0.0147 (Z) Otolaryngology (-) **, *p* = 0.0457 (Z)
**Parameters**	**Section 4** **UMS group vs. PMS group**	***p*-value (Statistical test)**
*Gender (M)*	35.35% < 42.36%	0.0003 * (C)
*University training*		
Preclinical	39.04% > 0.00%	<0.0001 * (C)
Clinical	61.96% < 100%	<0.0001 * (C)
*TB Vaccination (yes)*	0.24% < 18.87%	<0.0001 * (C)
*Positive Mantoux and* QFT-IT	0.62% < 2.08%	<0.0001 * (C)
**Parameters**	**Section 5** **Students with LTBI:** **UMS group vs. PMS group**	***p*-value (Statistical test)**
*Number*	US = 13; PMS = 18	
*Gender (M)*	30.77% < 38.89%	0.93 (CY)
*University training*		
Preclinical	46.15% > 0.0%	0.0060 * (CY)
Clinical	53.85% < 100%	0.0060 * (CY)
*TB Vaccination (yes)*	7.69% < 66.67%	0.001 * (C)

* = significant test, ** = most frequent; C = χ^2^ test; Cm = Multiple comparison χ^2^ test; Z = post hoc Z-test (only if multiple comparison χ^2^ test is significant); B = exact Binomial test; CY = χ^2^ test with Yates correction; T = t-Student Test for unpaired data; + = we considered in multivariate analysis only modalities with percentages different to zero; QFT-IT test = Quantiferon test; (+) = Quantiferon test was performed on subjects with positive Mantoux test; UMS = Undergraduate Medical Student; PMS = Postgraduate Medical Student.

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
