# Peer review of "Latent Tuberculosis Infection among Healthcare Students and Postgraduates in a Mediterranean Italian Area: What Correlation with Work Exposure?"

_ijerph, 2019, doi:10.3390/ijerph17010137_

Round 1

Reviewer 1 Report

Article:  Tuberculosis Infection among healthcare students and postgraduates in a Mediterranean Italian area: what correlation with work-exposure?

Tuberculosis is a current and very important topic. Healthcare workers (HCWs) are at a very high risk of latent TB infection (LTBI) and TB disease.  I have read this paper carefully, and found that it could be acceptable for publishing after absolving some major observations

Lines 51-53: The authors state “Tuberculosis is often a cause of hospitalization because most of the migrants come from areas with a high circulation of mycobacterium tuberculosis.” Although both premises are true, there is no clear causal relationship between them. So, it would be better no to use the connector “because”. I suggest changing this sentence for another one that reflects better the authors’ intentions. For example: “Tuberculosis is a frequent cause of hospitalization among the migrants coming from areas with a high circulation of mycobacterium tuberculosis”.

Lines 153-154: The authors write IFN-γ concentration level is not correlated with stage or degree of infection, immunological response level, or the probability of disease progression” This affirmation is questionable. LTBI is associated with higher circulating concentrations of interferon- γ (compared with non-LTBI), even after adjusting for age, gender, race, diabetes, hypertension, tobacco use, HIV status, body mass index, lipid profile, and lymphocyte count (1,2). In addition, according to Auld SC et al. patients with a positive tuberculin test (TST) have lower odds of disseminated disease (i.e., miliary or combined pulmonary and extrapulmonary disease), but there is no difference in the odds of disseminated disease with a positive IFN-γ release assay (IGRA). However, persons who are positive to either of these tests had lower odds of death. An indeterminate IGRA result is associated with greater odds of both disseminated disease and death (3). Furthermore, IGRA results are influenced by the level of immunosuppression, for example HIV status (4).

Lines 325-328: The authors state “In this regard, it is interesting to observe how some vaccinated subjects in our sample are infected with Mycobacterium tuberculosis: (MSS: 7.69 vs. PMS: 66.67%). This confirms that BCG vaccination is not very effective in preventing Mycobacterium Tuberculosis infection”. The fact that the proportion of MSS who are reactive with IGRA is higher than the proportion of PMS who are reactive with IGRA does not support the conclusion that BCG vaccination is not very effective in preventing Mycobacterium Tuberculosis infection —even though, according to the medical literature this affirmation could be true—. This, just could reflect selection bias, because a cross-sectional study like this is not the best design to stablish a causal association.

Lines 384-385: In Limitations of the study, the authors say “This study was a very exhaustive study, with a good sample size, but it was realized on a single geographical area”. There are well known risk factors, mainly HIV coinfection, which could interfere with TST and IGRA. Therefore, it would be appropriate to know Healthcare workers (HCWs) HIV status. If HIV status could not be tested, for example because of religious reasons, it must be clearly explained in Limitation of the study (5).

References:

Huaman MA, Deepe GS Jr, Fichtenbaum CJ. Elevated Circulating Concentrations of Interferon-Gamma in Latent Tuberculosis Infection. Pathog Immun. 2016 Fall-Winter;1(2):291-303.

Huaman MA, Henson D, Rondan PL, Ticona E, Miranda G, Kryscio RJ, et al. Latent tuberculosis infection is associated with increased unstimulated levels of interferon-gamma in Lima, Peru. PLoS One. 2018; 13(9):e0202191. doi: 10.1371/journal.pone.0202191.

Auld SC, Lee SH, Click ES, Miramontes R, Day CL, Gandhi NR, Heilig CM. IFN-γ Release Assay Result Is Associated with Disease Site and Death in Active Tuberculosis. Ann Am Thorac Soc. 2016; 13(12):2151-2158. DOI: 10.1513/AnnalsATS.201606-482OC

Latorre I, Martínez-Lacasa X, Font R, et al.IFN-γ response on T-cell based assays in HIV-infected patients for detection of tuberculosis infection. BMC Infect Dis 10, 348 (2010) doi:10.1186/1471-2334-10-348

Janagond AB, Ganesan V, Vijay Kumar GS, Ramesh A, Anand P,  Screening of health-care workers for latent tuberculosis infection in a Tertiary Care Hospital.Int J Mycobacteriol. 2017; 6(3):253-257. doi: 10.4103/ijmy.ijmy_82_17.

Author Response

Answers to revisor#1 comments in attachment

Reviewer 2 Report

The aim of this paper is to evaluate latent tuberculosis infection (LTBI) in undergraduate and postgraduate students to assess TB infection risk for health care workers/students in Palermo Italy.

The study identifies prevalence of TB in the groups studied and assesses professional risk by examining LTBI rates in groups according to clinical experience; course type or medical specialisation.

The strengths of the study include the large sample size of students tested and the detailed analysis of data.

Weaknesses of the study include results based on a "point in time" test (no serial screening); limited inclusion of/adjustment for other variables for exposure to TB in infected subjects (eg overseas travel; undertaking cough-inducing procedures; professional history; age). The small number of infected subjects limits generalisability of the results.

This study could provide a valuable contribution to the evidence base for HCW TB screening and surveillance for low TB incidence countries.

To realise this contribution I believe the manuscript requires major revision with considerable improvement of the discussion section so there is cohesion between the findings, the authors' interpretation of the findings and connection to the literature. Limitations of the study also need to be discussed.

The following feedback is provided to assist with strengthening the study/manuscript.

Abstract/Results

Lines 28 Keep consistency in describing subjects. It changed to "medical students" and "residents" where previously it was "undergraduates" and "postgraduates"

[later the names MSS and PMS, and then clinical and preclinical are used - chose one naming convention and stick with it throughout]

Line 29 The sentence starting "Considering the subjects previously vaccinated for TB...." has lost meaning when translated into English.

Introduction

Line 49 It is important to note that identifying LTBI will not eliminate TB unless it is treated.

Line 53 Please state this is an Italy-specific statement (or provide a reference if you are generalising). Australia, for example, has embraced use of IGRA and it is used exclusively for LTBI testing in some jurisdictions.

Line 54 Is there a reference for this statement? Such as the degree of risk (infection rate; case rate) suggests a group should be targeted for screening for LTBI.

Line 56 It is not clear if you are referring to HCWs having an increased risk of acquiring LTBI or active TB. Assuming you mean active TB (because that is where LTBI leads) then the contribution of HCWs to Italy's TB notifications would provide context for this study. With regard to LTBI, is the prevalence of LTBI in the Italian-born population known? If so, how does the prevalence of TB in this HCW cohort compare? Could the prevalence of LTBI in these HCWs be representative of an age-adjusted community infection rate and not be HC-related?

Line 64 I interpret your study to be the comparison of LTBI prevalence between two HCW student groups with different clinical exposure times. I do not see any correlation between "surveillance study on the usefulness of TST and QFT-IT to evaluate positive subjects...." and the stated aim, and what you report as findings, discussion and conclusions.

[Serial TB screening is usually used for HCW TB surveillance - why you have used a different surveillance method (and any limitations this presents) should be considered and discussed at some point in the paper]

Line 74 A brief rationale for using QFT-IT for confirming a positive TST should be provided (you are privileging the QFT-IT result), especially since it has been stated earlier that neither test is a gold standard for diagnosing LTBI.

A point to consider to add depth to the discussion:

The risk posed by HCWs with active TB to vulnerable patients is an important aspect for a HCW TB screening program. As the demographics of Italy changes it is likely people from high TB incidence countries join the workforce. What do the findings offer for this scenario?

Methods

Line 103 Is the word "prior" missing?...."informed consent letter prior to the processing of data".

Line 103 Can the name of the Ethics Committee be included please.

Line 107 The abbreviation TBC has not be defined.

Line 119 The greater specificity of IGRA needs to be explained here or in the introduction (see line 74 comment).

Results sub-heading missing

English translation and formatting made this section difficult to read. The tables were very helpful and mostly formatted well.

Table 1

1) Students with LTBI (TST and QFT-IT positive) as a proportion of all MSS students should be included (this is reported in the abstract) rather than only reporting QFT-IT positive as a proportion of TST positive students.

2) Nationality - other (4) is inconsistent with the text (6).

Table 2

Students with LTBI as a proportion of all PMS students should be included rather than only reporting QFT-IT positive as a proportion of TST positive students.

Lines 157-163 This is very difficult to follow due to the inconsistency in naming the two groups and starting sentences with percentages.

Line 175 Inconsistency in number of foreign born students (#8) with the tables (#6).

Line 182 Abbreviation TBC not defined.

Line 1986 I suggest numbering the sections of table 4 to correspond with the text.

196 The naming of groups has changed again. They are now referred to as  "clinical" and "preclinical". I cannot ascertain what is being reported here.

Can the total number of students (MSS+PMS) with LTBI be reported? I think the overall low rate of LTBI in this cohort of Italian HCWs is a useful finding for publication.

Line 202 Is there a reason the p values are not reported here?

Line 210 (and Line 250) The statement "lower presence of residents without LTBI" is awkward. Suggest stating the schools with subjects with LTBI are .....

Line 221 I suggest the results in section 1 of table 4 for the variable University training for the PMS group are not useful as all the PMS subjects are all in the clinical group. Similarly the comparison of students with LTBI MMS vs PMS using University training as the variable is affected by all PMS 100% of PMS being clinical. Comparison of MSS and PMS students with LTBI vs MMS and PMS studnets without LTBI with University training (clinical/preclinical) as the variable (not by subgroup MMS/PMS) would be useful to support the authors' discussion/conclusion (unless the authors wish to limit the findings and conclusion to TB risks for undergraduate students).

Line 227 "higher work exposure" could be more explicit eg with a higher risk of exposure to patients with active TB in the workplace.

Discussion

This section needs a complete re-write.

The discussion should start with a brief summary of the main findings of the study, followed by a discussion of each of the main findings in light of the literature and the relevance of the finding to the study aim.

Some issues:

Line 236 should the word "relevant" be "significant"?

Line 241 It is stated students who had BCG due to MDR TB exposure were excluded from the TB Service - does this mean excluded from the study? If so this should be  be reported in the methods.

Line 244 It is inappropriate to ascribe causality to the results. It is not appropriate to state clinical training caused a higher incidence ....

Line 247 Why are positive Mantoux tests being discussed? when LTBI was defined as Mantoux and IGRA positive for this study.

Line 253 QFT-IT analysis has reduced the number of students interpreted as having LTBI. The analysis cannot actually change a student's infection and some of those with positive TST and negative IGRA may have LTBI - there is no way of knowing (unless they get active TB) as there is no gold standard test for LTBI.

Line 256 It is very ambitious to state this study confirms that BCG vaccination is not very effective in preventing MTB infection when there is no gold standard test for diagnosing LTBI; the numbers infected in this study are very small; and none of the infected subjects knew how and when they were infected which decreases the probability that the positive test is a true positive. I agree there is plenty of evidence supporting BCG being mostly good at preventing TB meningitis in children under 5 years......but this study is not aimed at assessing the efficacy of BCG.

Line 262 Statement about age not discussed in light of the literature and the relevance of the finding to the aim of the study.

Line 264-266 This explains why there are more women in the sample but it does not discuss why more women may have been infected in the MSS group.

Line 267/268 An unsupported statement.

Line 269 This sentence seems to fit with section starting line 252.

Line 273 Another ambitious statement to make without any behavioural/knowledge analysis in this study to support the assumption. Also the prevalence of LTBI (note not conversion identified through serial testing) is determined by extremely small number of subjects with LTBI.Eg Digestive surgery and emergency surgery are 50% 1 out of 2 subjects infected.

Further, variables such as work history in other specialty roles/facilities/countries; and personal history such as travel to high TB incidence countries do not seem to have been considered.

LTBI in subjects in Pediatrics and Legal Medicine suggests to me that the possibility of infection outside the workplace has to be considered as possible even in a low TB-incidence country.

The discussion focuses only on student behaviours when there is opportunity here to discuss the many aspects on an infection control program for preventing TB transmission in HC facilities.

Line 279 This is another ambitious statement that is not supported by references. While the number of notifications may increase, the proportion of cases in Italian-born population may drop and infection of the Italian-born population may not increase - depending on the strength of other aspects of the Italian TB prevention and control program (eg case detection, treatment completion, treatment access, universal social support, refugee health programs).

Conclusion

New information should not be presented in the conclusion (this belongs in the discussion).

Limitations

There are limitations with using a single test rather than serial screening to determine an infection timeline to assist with identifying likely location of infection.

Identification of and adjustment for other variables for TB infection risk has not be undertaken.

Lack of a gold standard diagnositic test for LTBI is a limitation.

Declarations

Can the Ethics Committee be named please.

Author Response

Answers to reviewer #2 comments in attachment

Round 2

Reviewer 1 Report

I am completely satisfied with authors´ replies. I have no further comments.

Reviewer 2 Report

The authors have provided satisfactory responses to the issues and questions I raised.

This manuscript is a resubmission of an earlier submission. The following is a list of the peer review reports and author responses from that submission.

Round 1

Reviewer 1 Report

I find this to be a very interesting paper. Mandatory TB vaccination has ceased in most countries, and this could lead to dire consequences if studies like these are not conducted. I applaud the authors for this work. 

Reviewer 2 Report

Dear authors,

Although your study may be of interest,  I am very concerned about some aspects related to text editing and writing; design, and execution of the research; analysis and interpretation of the results and finally in the conclusions obtained.

General comments

Dear authors,

It is essential to keep in mind that there is no gold standard test for the diagnosis of LTBI. Moreover, the results of the diagnostic tests are subrogated estimation of the LTBI. We should also remember that the term "prevalence" refers to the LTBI and the "proportions" to the test results. I have observed some confusion regarding these terms, and I strongly recommend revising the entire text.

On the other hand, in my personal opinion, statistical methods should be rethought, and the analysis of the data should be reanalyzed, as they could lead to wrong results and then wrong conclusions.

I also want to recommend that the conclusions of the study be strictly related only to the results of your study.

Abstract

Line 27 and 28: percentage TST repeated
Line 29-31: it is difficult to understand what you are explaining about vaccinations

Introduction

Line 45-47: Does this sentence have any relation to the aim of the study or the possible results?
Line 47: where it says "mycobacterium tuberculosis" you should put Mycobacterium tuberculosis
Line 48: it is necessary to put a bibliographic reference.
Lines 66-69: the use of hyphens may not be the best way to express different ideas in an article. This may be acceptable in other scientific contexts such as slides and posters.

Line 78-80: prevalence refers to a disease or a clinical state. It could be that the expression "prevalence of positivity to tuberculosis infection" was not the best.

Materials and Methods

Sampling and eligibility
Line 86-87: I recommend rewriting the inclusion criteria without numbers.

In your study, were there any exclusion criteria?
What was the total number of population that could participate?
Did the people who did not participate have similar characteristics to those studied?
Could there be any selection bias in your study?

Lines 92-96: this paragraph belongs to the results section, it would be better to change its location

Lines 104-106: there are several acronyms for tuberculosis. I recommend you review the entire text.

Methods

Line 110: how much time elapsed between TST and Quantiferon in TST + cases? This time is essential to know because recent previous TSTs may influence the results of the QFT as some studies have related. This fact should be at least mentioned in the discussion.

Lines 117-131: QFT is a well-known technique. In my opinion, such a long explanation is not necessary; you can reference some other paper and leave only the key points of the test.

Statistical methods
I strongly recommend deeply reviewing the statistical methods used.
And you are also considering the use of multivariate statistics.

RESULTS
There is no title of "results."

Tables must be self-explanatory.

In table 1, there are errors in the percentages in the nationalities and incidences of countries. I suggest you review all the tables.

In table 5: Comparison of proportions should take into account the characteristics of the baseline group. For example, if there are more women in the basal population, we could erroneously interpret that LTBI is more frequent in women.

There are problems in the text format of the following texts:
lines 151-159; lines 160-169 and lines 172-177.

DISCUSSION & CONCLUSIONS

I recommend adding a study limitations section.

I recommend that the conclusions of the study be strictly related only to the results of your study.

Regards

Reviewer 3 Report

The article "Prevalence of latent tuberculosis infection among healthcare students and post graduates in a Mediterranean Italian are: what correlations with work-exposure?" attempts to evaluate the prevalence of latent tuberculosis (LTBI) among students at the School of Medicine in Palermo, Italy.

Results show that 0.6% of undergraduates and 2.1% of graduate students were diagnosed with LTBI.

The authors conclude that the frequency of LTBI is low in the population studied.

The article has some significant flaws.

Problematic/unclear testing protocols: The authors define LTBI as a positive tuberculin skin test (TST) followed by a confirmatory positive Quantiferon (QFT). Positive TST (size of reaction in mm), however, is not defined. More importantly, the TST can cause false positive QFT when QFT immediately follows TST. According to the manufacturer QFT should be deferred until 6 months after TST placement to avoid cross reactivity. This cross reactivity makes it difficultly to interpret the finding that more graduate students were diagnosed with LTBI because this group also had higher frequency of BCG immunization, potentially leading to higher frequency of TST positivity and QFT positivity (due to cross reactivity). In other words, it is possible that a person who is BCG immunized, then has a positive TST, followed by a positive QFT may simply be a person with a positive TST because of BCG immunization and a positive QFT because of cross reaction with a recent TST. Authors should describe the time between TST and QFT and grapple with the possibility of cross reactivity as the reason for their results. Describing differences between groups: The between group comparisons in the paper are poorly formatted and difficult to understand. For example, the statement “Regarding country incidence of TB, in the MSS group we identified a significant presence of Mantoux-negative subjects from countries with a low TB incidence compared to subjects with latent tuberculosis (99.90% >92.3%)” doesn’t have much meaning. A more straightforward comparison would be an evaluation of positive TST (or QFT or LTBI) in patients from countries with a low TB incidence compared to patients from countries with a high TB incidence. Authors should fully reformat all between-group comparisons to make them simpler and easier to understand. Unsupported conclusions. The authors state that the frequency of LTBI is “low” without offering a reference group for comparison. The authors statement that the risk of LTBI in students is “similar” to other health workers has the same problem: no specific estimate in the comparison group by which to make their claim. Authors should include some statement about the frequency of in similar groups of subjects (students) from other low burden countries in the region (i.e., Spain, Greece) to contextualize their findings.

Given these flaws, the manuscript has little validity as a scientific paper and does not add to our understanding of the TB epidemic as the authors intend.